# Sudden Unexpected Infant and Perinatal Death: Pathological Findings of the Cardiac Conduction System [note 1]

**DOI:** 10.3390/diagnostics15131637

**Published:** 2025-06-27

**Authors:** Giulia Ottaviani, Patrizia Leonardi, Massimo Del Fabbro, Simone G. Ramos

**Affiliations:** 1Lino Rossi Research Center for the Study, Prevention of Unexpected Perinatal Death, SIDS, Anatomic Pathology, Department of Biomedical, Surgical and Dental Sciences, Università Degli Studi Di Milano, 20122 Milan, Italy; patrizia.leonardi@unimi.it; 2Department of Biomedical, Surgical and Dental Sciences, Università Degli Studi Di Milano, 20122 Milan, Italy; massimo.delfabbro@unimi.it; 3Fondazione IRCCS Ca’ Granda, Ospedale Maggiore Policlinico, 20122 Milan, Italy; 4Pathology and Legal Medicine, Ribeirão Preto Medical School, University of São Paulo, Ribeirão Preto 14040-900, Brazil; sgramos@fmrp.usp.br

**Keywords:** sudden infant death syndrome (SIDS), sudden neonatal unexpected death, sudden intrauterine unexpected death (SIUD), cardiac conduction system, post-mortem examination

## Abstract

**Objective:** Sudden infant death syndrome (SIDS), sudden neonatal unexpected death (SNUD), and sudden intrauterine unexpected death (SIUD) are major unsolved, shocking forms of death that occur frequently and without warning. The body of literature on the anatomo-pathological substrates in the cardiac conduction system of SIDS-SIUD and their possible relationship with risk factors and triggers is fragmentary and scarce. The work aims is to analyze the cardiac conduction system findings collected at the national referral center for SIDS-SIUD. **Methods:** A total of 123 autopsied cases of SIDS (59.35% males, 40.65% females, mean age ± SD: 103.49 ± 67.17 days), 36 cases of SNUD (61.11% males, 38.89% females, mean age ± SD: 8.4 ± 9.17 days), and 127 cases of SIUD (45.67% males, 54.33% females, mean age ± SD: 36 ± 4.59 gestational weeks) were analyzed. In-depth pathological examinations of the cardiac conduction system were performed on serial sections according to the Lino Rossi Research Center’s protocol. **Results:** Among the studied cases, the following findings were observed: resorptive degeneration (SIDS: 88.7%, SNUD: 88.88%, SIUD: 56.69%), fetal dispersion (SIDS: 73.17%, SNUD: 91.66%, SIUD: 78.74%), Mahaim fibers (SIDS: 40.65%, SNUD: 44.44%, SIUD: 32.28%), cartilaginous meta-hyperplasia (SIDS: 56.91%, SNUD: 25%, SIUD: 33.07%), septated atrio-ventricular junction (AVJ) (SIDS: 21.14%, SNUD: 33.33%, SIUD: 38.58%), AVJ duplicity (SIDS: 6.5%, SNUD: 11.11%, SIUD: 2.36%), intramural bifurcation (SIDS: 3.25%, SNUD: 2.77%, SIUD: 4.72%). **Conclusions:** The prevalence of cardiac conduction findings was consistent across the SIDS, SNUD and SIUD groups. These findings provide valuable insights into the pathological characteristics of the cardiac conduction system in SIDS-SIUD that are potential morphological substrates for the development of cardiac arrhythmias. Further investigation and study of the conduction system are needed to understand the underlying mechanisms of these forms of death.

## 1. Introduction

Sudden Infant Death Syndrome (SIDS), also known as crib death, is defined as the sudden unexpected death of an infant within one year of age, where the fatal episode apparently occurs during sleep and remains unexplained after a thorough investigation, including a complete autopsy and review of the circumstances of death and clinical history [1]. SIDS represents the most prevalent form of death among infants aged between one month and one year. With an incidence of 0.38 per 1000 live births [2], SIDS remains a devastating occurrence wherein an apparently healthy infant experiences sudden and unanticipated demise. Consequently, the topic holds immense significance within the medical and public spheres, prompting heightened attention and research endeavors.

Since the definition of SIDS by Willinger et al. in 1991 [3], more than three decades of SIDS research have unveiled only fragments of its anatomo-pathological substrates.

Sudden Intrauterine Unexpected Death (SIUD), also known as unexpected stillbirth, refers to the death of a fetus at or after 25 weeks of gestation that occurs prior to complete delivery or extraction from the mother. This death is unforeseen based on the clinical history and remains unexplained even after a thorough review of the maternal history and a complete autopsy of the fetus, which includes an examination of the placental disk, umbilical cord, and membranes [4]. SIUD occurs at a rate up to ten times higher than SIDS, and its incidence has remained relatively unchanged over the past three decades, despite significant advancements in maternal and infant healthcare [5].

Sudden Neonatal Unexpected Death (SNUD) is the death of a newborn, aged from birth to one month, sudden, unexpected by history, and unexplained after a thorough case investigation, including performance of a general autopsy, examination of the death scene, and review of the clinical history [4]. SNUD can be the result of a Sudden Unexpected Postnatal Collapse (SUPC), which has an incidence of 0.05 per 1000 live births [6]. Recent forensic case reports further contribute to the understanding of SUPC [7].

The SIDS-SIUD complex [4] includes the three pathological diagnoses of SIDS, SIUD, and SNUD as a unified form of death, considered part of the same pathology. However, this terminology is scarcely used and the research on the cardiac conduction system for the three age groups SIDS, SIUD, and SNUD are scarce and fragmentary. The work aims to analyze the cardiac conduction system findings collected at the Italian national referral center for SIDS-SIUD. Preliminary findings have been reported in abstract form [8].

## 2. Materials and Methods

### 2.1. Selection and Classification of Cases

A total of 286 consecutive autopsied cases of fetuses, newborns or infants collected at the Lino Rossi Research Center for the study and prevention of unexpected perinatal death and Sudden Infant Death Syndrome (SIDS), Department of Biomedical, Surgical and Dental Sciences, Università degli Studi di Milano, Milan, Italy, were selected for this study.

The legal guardians of the deceased fetuses, newborns, or infants granted written informed consent for the autopsy and the research on the collected organs and tissues, in compliance with Italian law no. 31 dated 2 February 2006, “Regulations for Diagnostic Post-Mortem Investigation in Victims of Sudden Infant Death Syndrome (SIDS) and Unexpected Fetal Death” [9]. The confidentiality and privacy of personal data in accordance with current Italian and European regulations were respected.

The institutional review board (IRB) at the Lino Rossi Research Center of the Department of Biomedical, Surgical and Dental Sciences, Università degli Studi di Milano, reviewed and approved the research protocol (protocol code 0001) on 30 October 2023.

Ethical consent was not required for our study as the Lino Rossi Research Center is the referral National center for the study of sudden unexpected and unexplained infant and perinatal death, according to the Italian Law no. 31 of 2-02-2006 “Regulations for Diagnostic Post Mortem Investigation in Victims of Sudden Infant Death Syndrome (SIDS) and Unexpected Fetal Death”. 

For this study, the sample included 286 consecutive cases of sudden unexplained death in infancy or before birth referred to the Lino Rossi Research Center at the Università degli Studi di Milano, after non-natural causes of death were ruled out and toxicological analyses returned negative for substances such as drugs or alcohol. The general autopsy had failed to determine the cause of death, prompting further in-depth anatomical and pathological investigations, with a particular focus on the cardiac conduction system and brainstem.

Based on the age at death, the cases were grouped post-mortem into three categories, as follows:Sudden Intrauterine Unexpected Death (SIUD) if death occurred before birth [4];Sudden Neonatal Unexpected Death (SNUD) if death occurred from birth to one month [4];Sudden Infant Death Syndrome (SIDS) if death occurred from one month to one year [1,4].

A case of SIUD, SNUD, or SIDS was classified as “gray zone” or borderline if the review of the clinical history and complete autopsy revealed an additional event—such as mild pneumonia or chorioamnionitis—that acted as a triggering factor in an otherwise vulnerable individual, even though that event alone would not have been sufficient to cause death [10].

Autopsy cases referred to the Lino Rossi Center were retrospectively included in the study only if the entire cardiac conduction system was available for analysis. Detailed pathological examinations of the cardiac conduction system were then carried out on serial sections following the protocol established by the Lino Rossi Research Center [10].

### 2.2. Necropsy Guidelines

For every case included in the study, clinico-pathological information regarding the performance of general autopsy, eventual symptoms, familial occurrence, electrocardiograms, and circumstances of death were provided by the referring centers and retrospectively reviewed. A comprehensive analysis was then performed following the necropsy protocols established by the Lino Rossi Research Center, Department of Anatomic Pathology, Università degli Studi di Milano, Milan, Italy, with a particular emphasis on analyzing the cardiac conduction system and the brainstem through serial sectioning [10].

At the referring centers, the general autopsy was carried out involving a thorough gross and microscopic evaluation of all organs; in fetal cases, this also included an examination of the fetal adnexa (placental disk, umbilical cord, and membranes). All specimens were fixed in 10% phosphate-buffered formalin, then processed and embedded in paraffin. At the Lino Rossi Center, each heart was examined for pathological alterations in the atria, septa, ventricles, pericardium, endocardium, and coronary arteries. The origins of the coronary arteries were inspected, and multiple samples from the two primary coronary arteries and their major branches were collected for analysis. Sections of the myocardium and coronary arteries were subsequently stained using Hematoxylin and Eosin (HE) and Trichromic Heidenhain (Azan).

For the morphological analysis of the cardiac conduction system, the entire heart was processed in small fetuses, whereas for term fetuses, newborns, and infants, two specific heart specimens were collected for paraffin embedding:The first specimen block included the sinoatrial node (SAN). The *sulcus-crista terminalis* was the landmark for specimen removal: two longitudinal cuts were made parallel to the *sulcus-crista terminalis* through the right atrial wall.The second specimen block included the atrioventricular junction (AVJ), i.e., the atrio-ventricular node (AVN), the His bundle (HB), the bifurcation of the His bundle, and the right and left bundle branches. The *pars membranacea septi* was the landmark for specimen removal. The *pars membranacea septi* was identified placing the opened heart against a source of light. Two longitudinal cuts were made parallel to the *pars membranacea septi* through the interventricular wall.

Both specimens were fixed in 10% buffered formalin and embedded in paraffin. Serial sections were cut at intervals of 20–40 μm, and from each level, three 8 μm sections were collected, mounted, and alternately stained with Hematoxylin and Eosin (HE) and Azan. All remaining sections were stored and stained as necessary [10].

### 2.3. Statistical Analysis

Data are presented as mean ± standard deviation (SD). Differences between groups were assessed using Student’s *t*-test, Chi-square test, or Fisher’s exact test. Relationships between variables were evaluated using Pearson’s correlation analysis. Statistical analyses were performed with SigmaStat^®^ (version 4, Systat Software Inc., Chicago, IL, USA), and graphs were generated using SigmaPlot^®^ (version 14, Systat Software Inc., Chicago, IL, USA). A two-tailed *p*-value of less than 0.05 was considered statistically significant.

## 3. Results

### 3.1. Demographic and Clinical Data

An in-depth examination of the cardiac conduction system was carried out in 286 cases of sudden unexpected perinatal and infant death of natural causes grouping the victims according to the age at death. The data on sex for the age groups are shown on Table 1.

In this study, a diagnosis of SIUD was established in 127 fetuses, 58 (45.67%) males and 69 (54.33%) females; age range: 22–41 gws (mean age ± SD, 36.01 ± 4.57 gws). The 36 SNUD newborns were 22 (61.11%) males and 14 (38.89%) females; age range: 1 h-26 days after birth (mean age ± SD, 8.41 ± 9.17 days). The 123 SIDS infants were 73 (59.35%) males and 50 (40.65%) females; age range: 30–360 postnatal days (mean age ± SD, 103.49 ± 67.17 days) (Table 1).

Across the three age-related groups, the sex distribution differed significantly. Overall, males were statistically more frequent than females (*p* < 0.05) (Figure 1) (Table 1).

A total of 10 cases of SIUD were classified as SIUD gray zone, due to concomitant chorioamnionitis (5 cases), Parvovirus (2 cases), left ventricular noncompaction cardiomyopathy (2 cases), and amniotic fluid aspiration (1 case). A total of 7 cases of SNUD were classified as SNUD gray zone, due to concomitant pneumonia (4 cases), amniotic fluid aspiration pneumopathy (2 cases), and hyaline membrane pneumopathy (1 case). Ten cases of SIDS were classified as SIDS gray zone due to concomitant diagnosis of pneumonia (5 cases), dysplasia of the pulmonary artery (3 cases), lymphocytic myocarditis (1 case), and malaria (1 case).

### 3.2. Cardiac Conduction System Findings

In a total of 286 cases—127 SIUD, 36 SNUD and 123 SIDS—the CCS has been fully analyzed, through the serial sections method.

Resorptive degeneration, cartilaginous meta-hyperplasia, septated AVJ, and AVJ hemorrhage were found to be distributed differently among SIUD, SNUD and SIDS (*p* < 0.05) (Table 1).

Resorptive degeneration was identified in 56.69% of SIUD cases, in 88.88% of SNUD and in 88.7% of SIDS cases. Statistical analysis revealed an age-related significant difference across the three groups (*p* < 0.05) (Table 1). Within these areas of resorptive degeneration, clusters of young fibroblasts actively depositing collagen were observed, often embedded within a central fibrous core. These fibroblast clusters were either isolated from or positioned adjacent to the conduction structures [10] (Figure 2). No inflammatory response, extensive necrosis, or hemorrhage was detected in these areas. Macrophages were occasionally present near small degenerative foci, likely functioning as scavenger cells.

Areas of fetal dispersions, such as AVJ tissue undergoing the process of resorptive degeneration (Figure 2), were observed in 73.17% of SIDS, 91.66% of SNUD, and in 67.71% of SIUD cases, without significant differences (Table 1).

Islands of conduction system in the CFB (Figure 3), sometimes undergoing the process of resorptive degeneration, were detected in 53.66% of SIDS, 52.77% of SNUD, and in 67.71% of SIUD, without significant differences (Table 1).

Mahaim fibers, as accessory pathways connecting the AVJ directly with the myocardium of the interventricular septum (Figure 3), were detected in 40.56% of SIDS, 44.44% of SNUD, and in 32.28% of SIUD cases, without significant differences (Table 1).

Cartilaginous meta-hyperplasia of the CFB (Figure 4) was detected more frequently in SIDS (56.91% of cases) than in SNUD (25%) and SIUD (33.07), with an age-related distribution (Table 1).

Septated AVJ, characterized by fibrous tissue infiltration from the CFB into the AVJ (Figure 4), was observed less frequently in the SIDS group (21.14%) than in the SNUD (33.33%) and SIUD (38.58%) with an age-related distribution (Table 1).

AVJ hemorrhage (Figure 5), a consequence of the resuscitation maneuvers of cardiac massage, was detected in 14.63% of SIDS, 13.88% of SNUD, and 2.36% of SIUD cases; these latest detected only in cases of intrapartum SIUD, with significant age-related differences (Table 1).

Dislocated bifurcation (Figure 6A) was detected in 8.93% of SIDS, 13.88% of SNUD, and in 11.23% of SIUD cases, without significant differences among groups (Table 1).

Fibromuscular thickening of the AVN artery, as fibromuscular dysplasia or as initial phase of pre-atherosclerotic lesion [11], has been detected in 8.13% of SIDS, 5.55% of SNUD, and 2.36% of SIUD cases, without significant differences among groups (Table 1).

AVJ duplicity, or split AVJ, was observed in 6.5% of SIDS, in 11.11%, and in 2.36% of SIUD, without significant differences among groups (Table 1).

Intramural right bundle branch (Figure 6B) was detected in 3.35% of SIDS, 2.77% of SNUD, and in 4.72% of SIUD cases, without significant differences among groups (Table 1).

Intramural left bundle branch was observed in 1.62% of SIDS, 8.33% of SNUD, and in 0.79% of SIUD cases, with statistically significant difference among groups (Table 1).

Hypoplasia of the AVJ was detected in 2.43% of SIDS, 5.55% of SNUD, and in 2.36% of SIUD cases. without statistically significant difference (Table 1).

Hypoplasia of the SAN was detected in 4.06% of SIDS, 2.77% of SNUD, and in 1.57% of SIUD cases. without statistically significant difference (Table 1).

Fibromuscular thickening of the sino-atrial node (SAN) artery, as fibromuscular dysplasia or as initial phase of pre-atherosclerotic lesion [11], has been detected in 3.25% of SIDS cases, 0% SNUD cases, and 3.15% of SIUD cases, without correlation with age-increase (Table 1).

Hypoplasia of the central fibrous body (CFB) was detected in 1.62% of SIDS, 2.77% of SNUD and in 3.94% of SIUD cases, without statistically significant difference (Table 1).

## 4. Discussion

The present study examined a cohort of cases comprising Sudden Infant Death Syndrome (SIDS), Sudden Neonatal Unexpected Death (SNUD), and Sudden Intrauterine Unexpected Death (SIUD) to evaluate the prevalence of various cardiac conduction system findings. The results demonstrated distinct patterns in the observed pathological features among the studied cases.

SIDS, SNUD, and SIUD refer to an unforeseen fatal event occurring in a fetus or infant who appeared to be in good health, making the sudden outcome entirely unpredictable. The emotional impact on affected families is profound, carrying significant social consequences, particularly due to the high incidence of post-traumatic stress disorders among family members [12,13].

Once unnatural causes of death are excluded by the medical examiner, little attention, if any, is typically directed toward identifying the underlying mechanisms of these forms of death in the cardiac conduction system, the core of the heart, where the heart rhythm arise and spreads.

The present study provides a comprehensive analysis of the CCS in cases of SIUD, SNUD, and SIDS. Our findings reveal distinct patterns of histopathological anomalies in the CCS across these groups, highlighting potential age-related differences in vulnerability to sudden death.

A key observation in our study is the high prevalence of resorptive degeneration in the CCS, particularly in SIDS (88.7%) and SNUD (88.88%), compared to SIUD (56.69%) (Figure 2, Table 1). Resorptive degeneration has been previously described as a degenerative process affecting the CCS and has been associated with electrical instability leading to fatal arrhythmias [10]. The significantly lower incidence in SIUD suggests that this process may become more pronounced postnatally, possibly due to environmental and developmental influences.

Fetal dispersion, another commonly observed anomaly, was found in 91.66% of SNUD cases, 78.74% of SIUD cases, and 73.17% of SIDS cases (Figure 2, Table 1). This histological feature is related to the process of resorptive degeneration as part of the normal process of AVJ shaping toward the adult features [10]. However, fetal dispersion represents a pattern of electrically instable heart. The high prevalence in all groups suggests that fetal dispersion may be a common underlying factor in unexplained sudden deaths at different developmental stages.

Islands of conduction system in the CFB (Figure 3), sometimes undergoing the process of resorptive degeneration, may represent the source of ectopic electrically instable foci of cardiac impulse [10], were detected in 53.66% of SIDS, 52.77% of SNUD and in 67.71% of SIUD, without significant differences (Table 1).

Mahaim fibers (Figure 3), accessory conduction pathways often associated with arrhythmogenic potential, were detected in 40.65% of SIDS cases, 44.44% of SNUD cases, and 32.28% of SIUD cases. They connect the AVJ directly to the interventricular septum, bypassing the normal impulse pathway. Their presence has been reported in previous studies on conduction system abnormalities in SIDS and SIUD [10]. The relatively high incidence in our study reinforces the hypothesis that these fibers may contribute to conduction disturbances and increase the risk of lethal arrhythmias

Cartilaginous meta-hyperplasia (Figure 4) was notably more frequent in SIDS (56.91%) than in SNUD (25%) or SIUD (33.07%) (Table 1). This anomaly has been described in relation to excessive fibrocartilaginous tissue deposition within the CCS [10], potentially leading to displacement of the AVJ stuctures (Figure 6A) and conduction block. The statistically significant age-related differences suggest that cartilaginous meta-hyperplasia may progress with postnatal development, increasing susceptibility to fatal arrhythmias in infancy.

Structural anomalies of the AVJ were also prevalent in our study. Septated AVJ (Figure 4) was observed in 38.58% of SIUD cases, 33.33% of SNUD cases, and 21.14% of SIDS cases. The significance of septated AVJ in sudden death remains under investigation, but it is proposed that anatomical disruptions at this critical conduction node could lead to conduction delay or block.

Fibromuscular thickening of the AVN or SAN artery, either as fibromuscular dysplasia or as an initial phase of a pre-atherosclerotic lesion, was identified in few cases (Table 1), without statistically significant differences among groups. Fibromuscular dysplasia has been implicated in vascular abnormalities leading to impaired blood supply to the CCS, potentially contributing to conduction disturbances and arrhythmic events [10]. The presence of this lesion at a young age raises concerns regarding its potential progression into atherosclerotic disease, which has been recognized as a contributor to adult cardiac arrhythmias and sudden cardiac death [10]. Although its role in sudden unexplained deaths remains speculative, the detection of fibromuscular thickening in these cases suggests that vascular changes within the AVN artery may be an underrecognized factor in CCS pathology.

Similarly, AVJ duplicity, though less frequent, was detected in 6.5% of SIDS cases, 11.11% of SNUD cases, and 2.36% of SIUD cases. The presence of dual conduction pathways may contribute to reentrant arrhythmias, a known mechanism in sudden cardiac death [10].

AVJ hemorrhage (Figure 5), a known consequence of resuscitation maneuvers such as cardiac massage [10], was detected in 14.63% of SIDS cases, 13.88% of SNUD cases, and 2.36% of SIUD cases, with the latter occurring exclusively in intrapartum SIUD cases. The statistically significant age-related differences suggest that AVJ hemorrhage may be more commonly associated with resuscitation attempts in postnatal cases, whereas its lower incidence in SIUD may be attributed to the absence of such interventions. Previous studies have highlighted AVJ hemorrhage as a consequence of aggressive resuscitation [10], complicating the differentiation between resuscitation-related injuries and primary pathological findings. Nonetheless, its presence warrants careful histopathological evaluation to distinguish resuscitation-induced changes from intrinsic pathological conditions contributing to sudden death.

Intramural bifurcation (Figure 6A), a less frequently observed anomaly, was noted in 3.25% of SIDS cases, 2.77% of SNUD cases, and 4.72% of SIUD cases. While its role in sudden death remains uncertain, intramural bifurcation has been suggested to influence electrical conduction pathways, potentially predisposing to arrhythmic events [10].

Intramural right bundle branch (Figure 6A) was detected in 3.35% of SIDS cases, 2.77% of SNUD cases, and 4.72% of SIUD cases, with no significant differences among the groups (Table 1). This anatomical variation has been described in relation to possible conduction delays and arrhythmic risk, particularly in individuals with underlying structural heart disease. While its clinical significance in sudden death remains unclear, previous studies [10] have suggested that the intramural location of conduction fibers may predispose them to ischemic or degenerative damage, which could contribute to electrical instability. The findings in our study support the need for further research to clarify the potential role of intramural right bundle branch anomalies in sudden unexplained deaths.

Intramural left bundle branch was observed in 1.62% of SIDS cases, 8.33% of SNUD cases, and 0.79% of SIUD cases, with statistically significant differences among the groups (Table 1). The presence of an intramural left bundle branch has been associated with delayed electrical conduction and an increased risk of arrhythmic events [10]. The significantly higher incidence in SNUD compared to the other groups suggests that this anatomical variation may have a greater impact during the neonatal period, potentially contributing to conduction instability and sudden death. Further studies are necessary to elucidate the functional consequences of this finding and its potential role in the pathogenesis of unexplained sudden death in infancy.

Hypoplasia of the AVJ was detected in 2.43% of SIDS cases, 5.55% of SNUD cases, and 2.36% of SIUD cases, without statistically significant differences among groups (Table 1). This anomaly has been reported also in adult hearts and could be linked to developmental defects in the conduction system, potentially leading to conduction block and arrhythmias [10]. While the absence of significant differences suggests that AVJ hypoplasia may not be a distinguishing factor among the studied groups, its presence remains relevant due to its potential impact on electrical conduction stability.

Hypoplasia of the SAN was detected in 4.06% of SIDS cases, 2.77% of SNUD cases, and 1.57% of SIUD cases, without statistically significant differences among the groups (Table 1). Although these differences are not statistically significant, the presence of SAN hypoplasia remains noteworthy given the pivotal role of the SAN as the primary pacemaker of the heart. Congenital underdevelopment of the SAN may lead to impaired automaticity and predispose individuals to sinus node dysfunction, potentially contributing to arrhythmic events and sudden death. Previous investigations have suggested that structural anomalies in the SAN could be implicated in the pathogenesis of SIDS and other unexplained sudden deaths [10]. Further research is warranted to better understand the functional impact of SAN hypoplasia and its interaction with other conduction system abnormalities in these vulnerable populations.

Hypoplasia of the CFB was detected in 1.62% of SIDS cases, 2.77% of SNUD cases, and 3.94% of SIUD cases, without statistically significant differences among the groups. The CFB plays a crucial role in providing structural support and electrical insulation between the atrial and interventricular myocardium. Although the observed prevalence of CFB hypoplasia is relatively low and does not differ significantly across the studied groups, its presence could potentially contribute to subtle conduction disturbances. Structural abnormalities in the CFB might impair its insulating function, possibly facilitating abnormal electrical communication between cardiac chambers and predisposing to arrhythmogenic events. While direct evidence linking CFB hypoplasia to sudden death remains limited, this finding warrants further investigation into its functional impact. Future studies that integrate detailed histopathological assessments with electrophysiological evaluations could shed more light on the role of CFB hypoplasia in the pathogenesis of unexplained sudden deaths.

Importantly, age-related statistically significant differences in the distribution of resorptive degeneration, cartilaginous meta-hyperplasia, septated AVJ, and AVJ hemorrhage (Table 1) suggest that developmental changes in the CCS may influence the risk of sudden death at different life stages. These findings support the hypothesis that SIDS, SNUD, and SIUD may share common pathogenic mechanisms related to CCS anomalies, but with variations in expression depending on the age of the affected individual. Our findings support the suggestion that SIDS, SNUD and SIUD are part of the SIDS-SIUD complex [4], as a continuum of the same form of death that can happen from fetal age through the completion of the first year of age.

These findings in the conduction system provide valuable insights into the pathological characteristics of the cardiac conduction system in cases of SIDS, SNUD, and SIUD. The results suggest potential morphological substrates that may contribute to the development of cardiac arrhythmias, highlighting the need for further investigation and study of the conduction system in understanding the underlying mechanisms of these tragic events.

## 5. Limitation of the Study

Without comparing findings in neonates and infants who died of known causes to healthy infants, it is difficult to determine whether the abnormalities found are specific to sudden deaths or may also be present in other populations, thus reducing their diagnostic specificity. Given the call for further investigation, the manuscript may present a vision more of fundamental research than of direct clinical application in the short term.

## 6. Conclusions

Overall, our study underscores the importance of detailed histopathological examination of the CCS in unexplained sudden deaths in utero and in infancy. Further research integrating molecular and genetic analyses is needed to elucidate the underlying mechanisms and improve risk stratification for these tragic events.

## Figures and Tables

**Figure 1 diagnostics-15-01637-f001:**
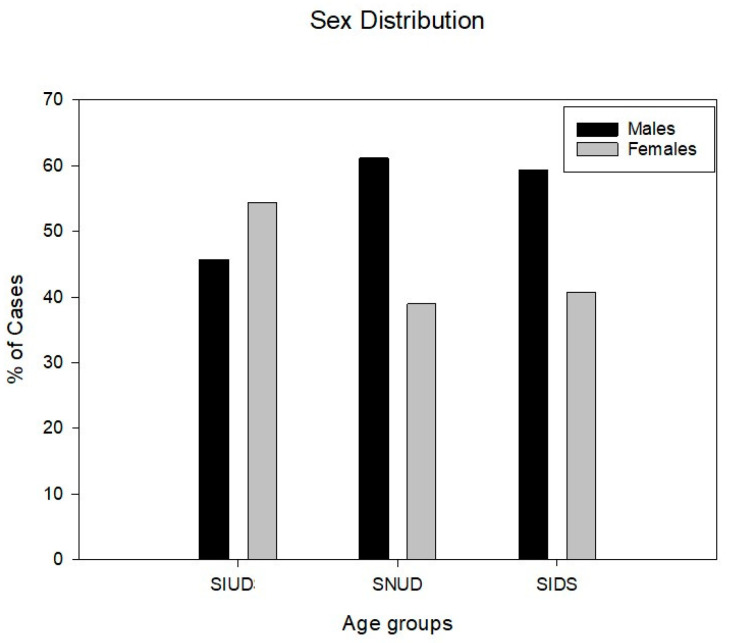
This graph shows the sex and the frequency distribution of the 286 studied cases, based on three forms of age-related death, SIUD—Sudden Intrauterine Unexpected Death, SNUD—Sudden Neonatal Unexpected Death; and SIDS—Sudden Infant Death Syndrome. The sex difference among groups is significantly different (*p* < 0.05).

**Figure 2 diagnostics-15-01637-f002:**
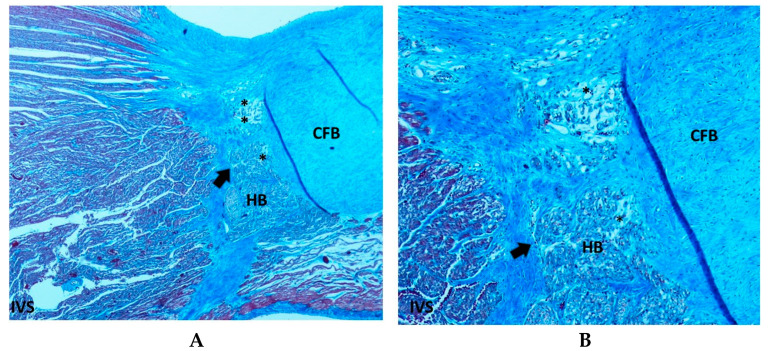
Section of the atrio-ventricular junction (AVJ) of a 39 + 6-week-gestation female fetus, victim of sudden intrauterine unexpected death (SIUD). The asterisks (*) point to the resorptive degeneration, embedded in the central fibrous body (CFB) and located in the peripheral areas of the His bundle (HB). An arrow points to a pattern of fetal dispersion of the HB. IVS = ventricular septum. Trichromic Heidenhain; Original magnification: (**A**) 40×, (**B**) 100×.

**Figure 3 diagnostics-15-01637-f003:**
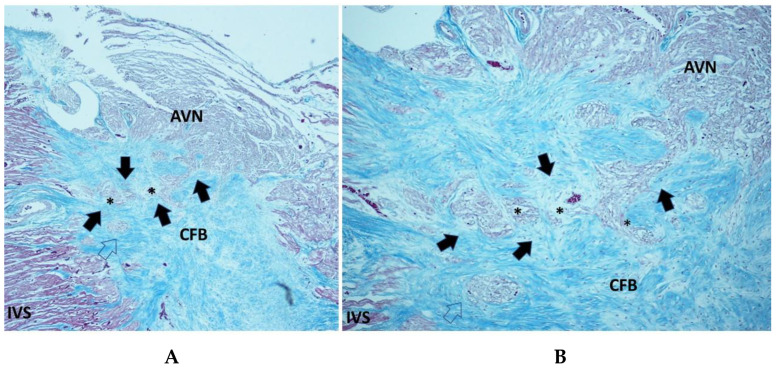
Section of the atrio-ventricular junction (AVJ) showing a Mahaim fiber (arrows), connecting the atrio-ventricular node (AVN) directly with the interventricular septum (IVS), in part undergoing the process of resorptive degeneration (*), in a 39-week-gestation female fetus, dying from sudden intrauterine unexpected death (SIUD). In the central fibrous body (CFB), there is an island of AVJ, separated from the AVN (empty arrow). Trichromic Heidenhain; Original magnification: (**A**) 40×, (**B**) 100×.

**Figure 4 diagnostics-15-01637-f004:**
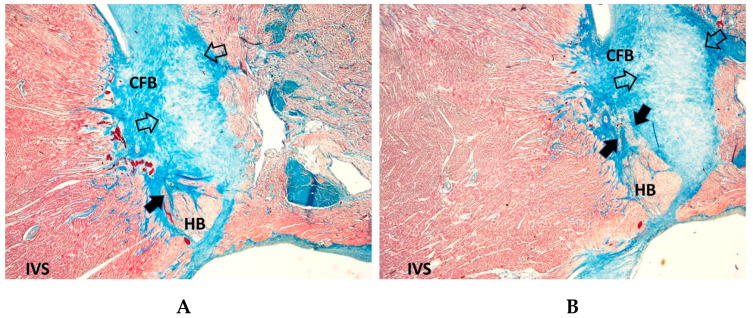
Cartilaginous meta-hyperplasia (empty arrows) of the central fibrous body (CFB), potentially compressing the His bundle (HB), in a 5-month-old boy who succumbed to Sudden Infant Death Syndrome (SIDS). The arrows point to a septated HB. IVS = interventricular septum. Trichromic Heidenhain; Original magnification: (**A**) 40×, (**B**) 100×.

**Figure 5 diagnostics-15-01637-f005:**
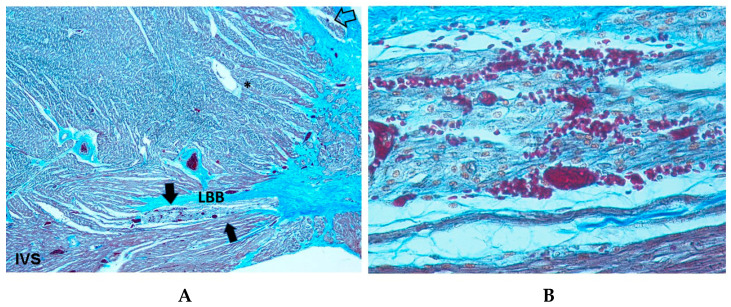
Section of the atrio-ventricular junction (AVJ) in a 17-day-old boy who died of Sudden Neonatal Unexpected Death (SNUD). (**A**) Arrows point to the hemorrhagic left bundle branch (LBB). The empty arrow points to the right bundle branch. (**B**) The hemorragic LBB is shown at higher magnification. IVS = interventricular septum. Trichromic Heidenhain; Original magnification: (**A**) 40×, (**B**) 400×. * The hemorrhage at higher magnification.

**Figure 6 diagnostics-15-01637-f006:**
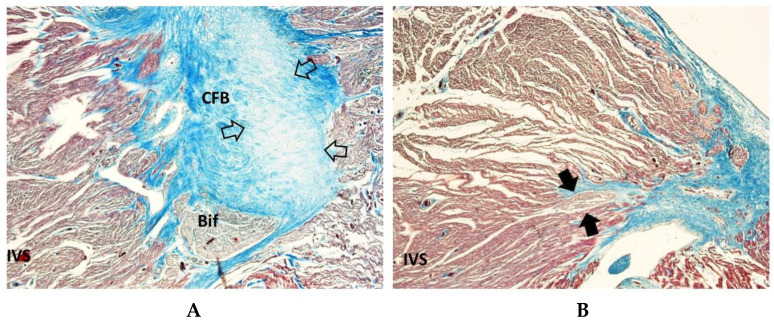
Consecutive sections of the atrioventricular junction of a 37 + 5 weeks’ gestation female fetus, who died from Sudden Intrauterine Unexpected Death (SIUD). The bifurcation of the His bundle (Bif) is dislocated, likely due to the cartilaginous meta-hyperplasia (empty arrows) of the central fibrous body (CFB). The right bundle branch (arrows) is intramural. IVS = interventricular septum. Trichromic Heidenhain; Original magnification: (**A**,**B**) 40×.

**Table 1 diagnostics-15-01637-t001:** Findings in the cardiac conduction system of the study cases.

Cardiac Conduction System	SIDS (*N* = 123) %	SNUD (*N* = 36) %	SIUD (*N* = 127) %	*p* Value
Gender (M/F) %	59.35/40.65	61.11/38.89	45.67/54.33	*p* < 0.05
Age range; mean ± SD	30–360 dy 103.49 ± 67.17	1 hr–26 dy 8.41 ± 9.17 dy	22–41 gw; 36.01 ± 4.57 gw	
Resorptive degeneration	88.7	88.88	56.69	*p* < 0.05
Fetal dispersion	73.17	91.66	78.74	
Islands in Central fibrous body	53.66	52.77	67.71	
Mahaim fibers	40.65	44.44	32.28	
Cartilaginous meta-hyperplasia	56.91	25	33.07	*p* < 0.05
Septated atrio-ventricular junction (AVJ)	21.14	33.33	38.58	*p* < 0.05
AVJ Hemorrhage	14.63	13.88	2.36	*p* < 0.05
Dislocated bifurcation	8.93	13.88	11.23	
Fibromuscular thickening AVN artery	8.13	5.55	2.36	
AVJ duplicity	6.5	11.11	2.36	
Intramural right bundle	3.25	2.77	4.72	
Intramural left bundle	1.62	8.33	0.79	*p* < 0.05
AVJ hypoplasia	2.43	5.55	2.36	
SAN hypoplasia	4.06	2.77	1.57	
Fibromuscular thickening SAN artery	3.25	0	3.15	
CFB (Central Fibrous Body) hypoplasia	1.62	2.77	3.94	

Age groups: Sudden Infant Death Syndrome (SIDS): 1–12 months; Sudden Neonatal Unexpected Death (SNUD): 0–30 days; Sudden Intrauterine Unexpected Death (SIUD): before birth. Abbreviations: M = male; F = female; gw = gestational weeks; *N* = Number of cases; hr = hour; dy = days; ms, months; SAN = sino-atrial node; HB = His bundle; AVN = atrio-ventricular node; AVJ = atrio-ventricular junction; BIF = bifurcation (of HB); CFB = central fibrous body; LBB = left bundle branch; RBB = right bundle branch. * Statistically significant; *p* < 0.05.

## Data Availability

The data presented in this study are available on request from the corresponding author. The data are not publicly available due to privacy and confidentiality of the data.

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
