# Peer review of "Sudden Unexpected Infant and Perinatal Death: Pathological Findings of the Cardiac Conduction System†"

_diagnostics, 2025, doi:10.3390/diagnostics15131637_

Round 1
Reviewer 1 Report
Comments and Suggestions for Authors
With this work, you have eliminated all the frustrations of doctors who cannot answer parents' questions about the reason for the early death of a child due to heart failure, whether in the intrauterine period, in the neonatal period and in infancy in both sexes. In all aspects of the development of pediatric cardiology and cardiac surgery, there has been a radical advance in diagnostic and therapeutic possibilities in the last 30 years, which has resulted in a decrease in perinatal mortality. This seems to be the first scientific work that shows the pathological morphology in sudden intrauterine death syndrome (SIUD), in the neonatal period (SNUD) or in infancy (SIDS). In a very large number of all these patients, including those with unexpected intrauterine death, a detailed and sophisticated pathohistological analysis of the cardiac conduction system in all three groups and in both sexes was performed, as well as an analysis of the coronary blood vessels, fetal organelles and the connection of the brain with the medulla oblongata (brainstem). In addition to the strict criteria of the overall histological analysis, the conduction system of the heart was specifically investigated (sinoatrial node, sulcus crista terminalis, atrioventricular node-junction (AVJ), bundle of His (HB), bifurcation of the bundle of His and pars membranacean septi. An extensive statistical analysis was performed with exact histological presentations of pathological changes in the mentioned segments of the specialized conduction system of the heart. Hemorrhagic changes in the AVJ system are important, intramural changes in the left bundle of His (LBBB). Resorptive degenerations in the AVN, IVS are shown, which primarily relate to SUID in terminal pregnancy, cartilaginous meta-hyperplasia in the CFB and AVJ, which primarily relates to SIDS), etc. These findings open a new chapter in the knowledge enabled by histopathological research and molecular-genetic analysis. I repeat, this work frees us from the suspicion that we made a mistake by not identifying the cause of sudden death in early childhood (newborn, infant), and even intrauterine. (I am grateful, because a month ago I experienced the sudden death of a child in terminal pregnancy, and the cause was not clear).

Author Response
- These findings open a new chapter in the knowledge enabled by histopathological research and molecular-genetic analysis. I repeat, this work frees us from the suspicion that we made a mistake by not identifying the cause of sudden death in early childhood (newborn, infant), and even intrauterine. (I am grateful, because a month ago I experienced the sudden death of a child in terminal pregnancy, and the cause was not clear).
Response. We thank the reviewer for the appreciation of our work

Reviewer 2 Report
Comments and Suggestions for Authors
The study is clear, straightforward, and innovative, and offers an important contribution to the forensic community. Here are some reviews
* Lack of Adequate Control Groups (Implicit): Although the abstract does not specify, a common criticism in descriptive pathology studies is the possible absence or insufficiency of comparable control groups. Without comparing findings in neonates and infants who died of known causes to healthy infants, it is difficult to determine whether the abnormalities found are specific to sudden deaths or may also be present in other populations, thus reducing their diagnostic specificity.
* Immediate Clinical and Translational Implications: Although the manuscript mentions the need for improved "risk stratification," the path to translate these pathological findings into concrete and immediately applicable clinical tools may not be fully delineated. Given the call for "further investigation," the manuscript may present a vision more of fundamental research than of direct clinical application in the short term.
Finally, the Authors should include a paragraph about the limitations of the study
Author Response
- * Lack of Adequate Control Groups (Implicit): Although the abstract does not specify, a common criticism in descriptive pathology studies is the possible absence or insufficiency of comparable control groups. Without comparing findings in neonates and infants who died of known causes to healthy infants, it is difficult to determine whether the abnormalities found are specific to sudden deaths or may also be present in other populations, thus reducing their diagnostic specificity.
Response. A limitation of the study section has been added to the manuscript
2.* Immediate Clinical and Translational Implications: Although the manuscript mentions the need for improved "risk stratification," the path to translate these pathological findings into concrete and immediately applicable clinical tools may not be fully delineated. Given the call for "further investigation," the manuscript may present a vision more of fundamental research than of direct clinical application in the short term.
Response. A limitation of the study section has been added to the manuscript:
“5. Limitation of the study
Without comparing findings in neonates and infants who died of known causes to healthy infants, it is difficult to determine whether the abnormalities found are specific to sudden deaths or may also be present in other populations, thus reducing their diagnostic specificity. Given the call for further investigation, the manuscript may present a vision more of fundamental research than of direct clinical application in the short term”.
[Manuscript page 13, lines 448-453]
- Finally, the Authors should include a paragraph about the limitations of the study.
Response. A limitation of the study section has been added to the manuscript
Round 2
Reviewer 2 Report
Comments and Suggestions for Authors
Here are some minor revisions for your paper, incorporating the new reference. Since I cannot directly edit the PDF, I will provide suggested changes and indicate where the new reference should be added.
Suggested Minor Revisions:
Here are some general minor revisions focusing on clarity, flow, and academic tone. These are examples and can be applied throughout the document:
General:
Standardize spacing: Ensure consistent spacing after punctuation (e.g., one space after a period).
Review for conciseness: Eliminate redundant words or phrases. For example, "The main objective of this work is to analyze..." could be "This work aims to analyze..."
Specific Examples for Introduction (Page 1-2):
Line 14 (Abstract Objective):
Original: "Objective Sudden infant death syndrome (SIDS), sudden neonatal unexpected death (SNUD), and sudden intrauterine unexpected death (SIUD) are major unsolved, shocking forms of death that occur frequently and can happen at any time without warning."
Suggested Revision: "Objective: Sudden Infant Death Syndrome (SIDS), Sudden Neonatal Unexpected Death (SNUD), and Sudden Intrauterine Unexpected Death (SIUD) are major, unsolved, and shocking forms of death that occur frequently and without warning."
Reason: Improves sentence flow and academic tone.
Line 41 (Introduction):
Original: "Sudden Infant Death Syndrome (SIDS), or crib death, is defined as the sudden unexpected death of an infant within one year of age, with onset of the fatal episode apparently occurring during sleep, that remains unexplained after a thorough investigation, including performance of a complete autopsy and review of the circumstances of death and the clinical history [1]."
Suggested Revision: "Sudden Infant Death Syndrome (SIDS), also known as crib death, is defined as the sudden, unexpected death of an infant within one year of age, where the fatal episode apparently occurs during sleep and remains unexplained after a thorough investigation, including a complete autopsy and review of the circumstances of death and clinical history [1]."
Reason: Improves clarity and sentence structure.
Line 66 (Introduction):
Original: "The SIDS-SIUD complex [4] include the three pathological diagnoses of SIDS, SIUD and SNUD as a unified form of death, being part of the same pathology."
Suggested Revision: "The SIDS-SIUD complex [4] includes the three pathological diagnoses of SIDS, SIUD, and SNUD as a unified form of death, considered part of the same pathology."
Reason: Corrects verb agreement and improves phrasing.
Integrating the New Reference:
The new reference is highly relevant to the discussion of "Sudden Unexpected Postnatal Collapse (SUPC)." You can integrate it into the Introduction section, specifically around where SUPC is first mentioned.
Proposed Integration in the Introduction (Page 2, around lines 63-64):
Original Sentence: "SNUD can be the result of a Sudden Unexpected Postnatal Collapse (SUPC) which as an incidence of 0.05 per 1,000 live births [6]."
Suggested Revision to include the new reference: "SNUD can be the result of a Sudden Unexpected Postnatal Collapse (SUPC), which has an incidence of 0.05 per 1,000 live births [6]. Recent forensic case reports further contribute to the understanding of SUPC."
You will need to renumber your existing references accordingly or add this as reference 13.
Add the following to your References section:
Esposito, M., Sessa, F., Nannola, C., Pignotti, M.S., Greco, P. and Salerno, M., 2024. Sudden unexpected postnatal collapse and BUB1B mutation: first forensic case report. International Journal of Legal Medicine, 138(5), pp.2049-2055.
Author Response
- Standardize spacing: Ensure consistent spacing after punctuation (e.g., one space after a period).
Response. We have ensured consistent spacing.
- Review for conciseness: Eliminate redundant words or phrases. For example, "The main objective of this work is to analyze..." could be "This work aims to analyze..."
Response. We have changed the sentences according to the reviewer’s comments
[Abstract line 23; Introduction line 74]
- Specific Examples for Introduction (Page 1-2):
Line 14 (Abstract Objective):
Original: "Objective Sudden infant death syndrome (SIDS), sudden neonatal unexpected death (SNUD), and sudden intrauterine unexpected death (SIUD) are major unsolved, shocking forms of death that occur frequently and can happen at any time without warning."
Suggested Revision: "Objective: Sudden Infant Death Syndrome (SIDS), Sudden Neonatal Unexpected Death (SNUD), and Sudden Intrauterine Unexpected Death (SIUD) are major, unsolved, and shocking forms of death that occur frequently and without warning."
Reason: Improves sentence flow and academic tone.
Response. We have changed the sentences according to the reviewer’s comments
[Abstract line 21[
- Line 41 (Introduction):
Original: "Sudden Infant Death Syndrome (SIDS), or crib death, is defined as the sudden unexpected death of an infant within one year of age, with onset of the fatal episode apparently occurring during sleep, that remains unexplained after a thorough investigation, including performance of a complete autopsy and review of the circumstances of death and the clinical history [1]."
Suggested Revision: "Sudden Infant Death Syndrome (SIDS), also known as crib death, is defined as the sudden, unexpected death of an infant within one year of age, where the fatal episode apparently occurs during sleep and remains unexplained after a thorough investigation, including a complete autopsy and review of the circumstances of death and clinical history [1]."
Reason: Improves clarity and sentence structure.
Response. We have changed the sentences according to the reviewer’s comments
[Introduction, lines 48-51[
- Line 66 (Introduction):
Original: "The SIDS-SIUD complex [4] include the three pathological diagnoses of SIDS, SIUD and SNUD as a unified form of death, being part of the same pathology."
Suggested Revision: "The SIDS-SIUD complex [4] includes the three pathological diagnoses of SIDS, SIUD, and SNUD as a unified form of death, considered part of the same pathology."
Reason: Corrects verb agreement and improves phrasing.
Response. We have changed the sentences according to the reviewer’s comments
[Introduction, lines 71-72[
- Integrating the New Reference:
The new reference is highly relevant to the discussion of "Sudden Unexpected Postnatal Collapse (SUPC)." You can integrate it into the Introduction section, specifically around where SUPC is first mentioned.
Proposed Integration in the Introduction (Page 2, around lines 63-64):
Original Sentence: "SNUD can be the result of a Sudden Unexpected Postnatal Collapse (SUPC) which as an incidence of 0.05 per 1,000 live births [6]."
Suggested Revision to include the new reference: "SNUD can be the result of a Sudden Unexpected Postnatal Collapse (SUPC), which has an incidence of 0.05 per 1,000 live births [6]. Recent forensic case reports further contribute to the understanding of SUPC."
You will need to renumber your existing references accordingly or add this as reference 13.
Add the following to your References section:
Esposito, M., Sessa, F., Nannola, C., Pignotti, M.S., Greco, P. and Salerno, M., 2024. Sudden unexpected postnatal collapse and BUB1B mutation: first forensic case report. International Journal of Legal Medicine, 138(5), pp.2049-2055.
Response. We have changed the sentence according to the reviewer’s comments
[Introduction, lines 70-71[
We have added the new reference # 7 and renumbered all the references
